# Computer-Aided Detection of Quantitative Signatures for Breast Fibroepithelial Tumors Using Label-Free Multi-Photon Imaging

**DOI:** 10.3390/molecules27103340

**Published:** 2022-05-23

**Authors:** Kana Kobayashi-Taguchi, Takashi Saitou, Yoshiaki Kamei, Akari Murakami, Kanako Nishiyama, Reina Aoki, Erina Kusakabe, Haruna Noda, Michiko Yamashita, Riko Kitazawa, Takeshi Imamura, Yasutsugu Takada

**Affiliations:** 1Department of Breast Center, Ehime University Hospital, Ehime 791-0204, Japan; taguchi.kana.kr@ehime-u.ac.jp (K.K.-T.); murakami.akari.yf@ehime-u.ac.jp (A.M.); nishiyama.kanako.ad@ehime-u.ac.jp (K.N.); aoki.reina.ik@ehime-u.ac.jp (R.A.); kusakabe.erina.iw@ehime-u.ac.jp (E.K.); noda.haruna.vo@ehime-u.ac.jp (H.N.); m-yamash@m.ehime-u.ac.jp (M.Y.); 2Department of Hepato-Biliary-Pancreatic Surgery and Breast Surgery, Ehime University, Ehime 791-0204, Japan; takaday@m.ehime-u.ac.jp; 3Department of Molecular Medicine for Pathogenesis, Graduate School of Medicine, Ehime University, Ehime 791-0204, Japan; timamura-ind@umin.ac.jp; 4Translational Research Center, Ehime University Hospital, Ehime 791-0204, Japan; 5Division of Diagnostic Pathology, Ehime University Hospital, Ehime 791-0204, Japan; riko@m.ehime-u.ac.jp

**Keywords:** breast fibroepithelial lesions, computer-aided diagnosis, deep learning, multi-photon microscopy, second harmonic generation

## Abstract

Fibroadenomas (FAs) and phyllodes tumors (PTs) are major benign breast tumors, pathologically classified as fibroepithelial tumors. Although the clinical management of PTs differs from FAs, distinction by core needle biopsy diagnoses is still challenging. Here, a combined technique of label-free imaging with multi-photon microscopy and artificial intelligence was applied to detect quantitative signatures that differentiate fibroepithelial lesions. Multi-photon excited autofluorescence and second harmonic generation (SHG) signals were detected in tissue sections. A pixel-wise semantic segmentation method using a deep learning framework was used to separate epithelial and stromal regions automatically. The epithelial to stromal area ratio and the collagen SHG signal strength were investigated for their ability to distinguish fibroepithelial lesions. An image segmentation analysis with a pixel-wise semantic segmentation framework using a deep convolutional neural network showed the accurate separation of epithelial and stromal regions. A further investigation, to determine if scoring the epithelial to stromal area ratio and the SHG signal strength within the stromal area could be a marker for differentiating fibroepithelial tumors, showed accurate classification. Therefore, molecular and morphological changes, detected through the assistance of computational and label-free multi-photon imaging techniques, enable us to propose quantitative signatures for epithelial and stromal alterations in breast tissues.

## 1. Introduction

Fibroepithelial tumors of the breast are common benign lesions consisting of epithelial and stromal components. These include fibroadenomas (FAs) and phyllodes tumors (PTs), commonly seen in clinical practice. FAs arise from the epithelium and stroma of the terminal duct lobular unit. The histological hallmark is a well-balanced epithelial and stromal proliferation. FAs are the most common benign breast lesions in young females. PTs were first fully characterized in 1838 by Müller [1], and are histologically characterized by epithelial-lined cleft-like spaces with hypercellular stroma, similar to a leaf in architecture. PTs include sub-classifications such as benign, borderline, and malignant, according to the WHO classification [2]. The distinction among the three sub-groups is based on the combination of several histologic features. Benign PTs sometimes relapse with higher proliferative activity, and may progress to a borderline PT or a malignant PT. Moreover, benign PTs mimic FAs in histopathology; therefore, both lesions are difficult to distinguish with a core needle biopsy (CNB) or vacuum-assisted biopsy (VAB), despite different clinical courses. Almost all patients diagnosed with FA are recommended for follow-up, but surgery may be considered when they exhibit rapid growth or a size greater than 3 cm in imaging. On the other hand, once a patient is diagnosed with a PT by biopsy, regardless of the size, a wide excision with surgical margins of ≥1 cm is recommended because of local recurrence [3]. Several studies have tried to find the clinical and histological factors to differentiate PTs from FAs, but these factors often overlapped. It was reported that the sensitivity of imaging and CNB for diagnosing PTs is 65% and 63% [4], and another report showed that the inter-observer variation was high with CNB when diagnosing fibroepithelial tumors [5]. Moreover, the upstaging to PT from fibroepithelial tumors diagnosed by CNB is often experienced in clinical practice. It was reported that the upstage rate was 37.5% for benign or borderline PTs in excised fibroepithelial tumors [6]. Stromal mitosis might be helpful for the differential diagnosis of a PT. Finding two or more stromal mitoses per 10 HPFs (high-power fields) in the CNB specimen may indicate a PT [7]. Immunohistochemical studies are also useful for the differential diagnosis of fibroepithelial tumors, such as Ki67 or topoisomerase II [7]. Nonetheless, a report showed mitotic counts, and these immunostaining findings on CNB materials overlapped between FAs and PTs [8]. Thus, there is no clear definable cut-off for these histological or immunohistochemical features. Tumor heterogeneities result in both the difficulty and the inter-observer variation in the diagnosis of fibroepithelial tumors. Therefore, although there are several predictive factors [3,9,10,11], these are still insufficient to recognize a distinction between benign PTs and FAs, and, hence, stand as a diagnostic challenge to decide clinical management [4,5,6].

A label-free imaging technique with multi-photon microscopy (MPM), which enables high-resolution fluorescence imaging, is attracting much attention as a histopathological diagnostic tool for assessing disease states. This technique has been extensively used for the assessment of various diseases, such as cancer [12,13], liver fibrosis [14,15], and osteoarthritis [16,17]. There are several applications to other tissues and diseases; see reviews [18,19,20]. The near-infrared beam used for multi-photon excitation can excite endogenous fluorophores, which include nicotinamide adenine dinucleotide (NADH), flavin adenine dinucleotide (FAD), elastic fibers, vitamins, and other metabolites [21]. In breast cancer tissues, this autofluorescence (AF) has been used to estimate cellular redox states, as well as for the assessment of cancers [22,23]. At the same time, second harmonic generation (SHG) imaging is possible through MPM, which allows the direct visualization of molecules possessing a non-centrosymmetric nature, such as collagen. SHG imaging plays vital roles in evaluating changes in fibrillar organization [24]. Combined SHG and AF imaging has been used for diagnosing breast cancers [22,23,25,26,27,28]. On the other hand, there have been few reported studies which surveyed the relation between benign breast lesions and MPM. Two sub-types of fibroadenomas were investigated using AF and SHG signals [29], and it was reported that the measurement of collagen density by SHG imaging was useful for the differential diagnosis of breast FAs and PTs [30]. The study simply investigated the amounts of collagen fibrils by quantifying the SHG signals as a potential diagnostic index. Although this achieved over 85% sensitivity and specificity, for more accurate detection, structural information regarding the collagenous density and proliferation of lactiferous ducts should be included, since fibroepithelial tumors are composed of the proliferation of stromal and epithelial elements. Therefore, in order to refine signatures that differentiate the lesions more precisely, we took advantage of a machine-learning-based approach for quantitative feature detection. Digital pathology is the field of computer-aided detection and evaluation of diseases, aiming to automate their assessment. With the advancement of artificial intelligence, this approach is becoming more common in clinical investigations. In this study, we aimed to find novel morphological signatures of FA and PT fibroepithelial lesions in breast tissues. To investigate whether morphological features which reflect stromal hypertrophy and epithelial proliferation can differentiate the lesions, we performed a pixel-wise semantic segmentation method using a deep learning framework. The accurate separation of epithelial and stromal regions allowed for the estimation of the balance of epithelial to stromal regions, which can be a signature for differentiating FAs and PTs. Furthermore, combining this with the collagen SHG signal strength led us to finding a refined index to distinguish FA and PT lesions.

## 2. Results

### 2.1. MPM Imaging Characterizes Morphological Distinctions between Epithelial and Stromal Regions for FA and PT Lesions

We first summarized the statistics of five FA and five PT patient samples, which were subjected to analyses (Table 1). The median sizes for FA and PT on the pre-operative imaging of ultrasound sonography were 3.0 cm (IQR 3.0–3.1 cm) and 2.9 cm (IQR 1.4–3.5 cm), respectively. For the biopsies, CNB or VAB were used. The median number of biopsied specimens was three in both the FA and PT groups. In the FA group, two patients (20%) could not be pre-operatively diagnosed with FA or PT lesions. On the other hand, in the PT group, two patients (20%) were pre-operatively diagnosed with FA. All patients underwent lumpectomy or mastectomy, and were finally diagnosed with FA or PT.

In order to examine how MPM images of SHG and AF signals characterize the morphological differences between epithelial and stromal regions in breast mammary gland tissues, we first performed an MPM observation and histological analyses of the tissue sections (Figure 1, Appendix A). The histological examination was performed using the HE and the PSR staining methods. Compared to the histological slices, the epithelial cellular structures were featured as slightly dark regions in the SHG images, and the boundary between the epithelia and stroma in the AF images could be recognized. On the areas corresponding to the epithelia, cell nuclei were stained in the HE slices, indicating that these included mammary duct epithelia and lumens. Collagen-rich stromal areas were recognized as strong SHG signal areas in MPM images, and regions stained red in the PSR staining images. Since PSR specifically stained collagen type I and III, the shapes and patterns of fibril structures were close to those observed in the SHG images, consistent with the SHG-illuminated collagen molecules. These results indicated that MPM images enabled us to morphologically differentiate epithelial and stromal tissues in breast tissues of fibroepithelial lesions.

### 2.2. Deep-Learning-Based Image Segmentation Approach for Differentiating Epithelial and Stromal Morphologies

It has been partly reported that PTs grow more in stromal regions than FAs [2]. Therefore, for establishing a quantitative criterion for differentiating FAs and PTs, we attempted to score the epithelial to stromal area ratio. Thus, for the automated quantification of these features, we took advantage of an image segmentation approach. In order to perform image segmentation, we employed SegNet, a deep convolutional neural network architecture for semantic segmentation. This deep learning-based framework was shown to be a high-performance architecture for generic scene semantic segmentation. Thus, to implement the supervised image segmentation, we first prepared ground-truth image sets, namely, labeled images with three types of categorized regions, ‘Epithelial’, ‘Stroma’, and ‘Outer’ regions (Figure 2). We manually selected these regions by comparing the MPM images with the HE- and PSR-stained images. ‘Epithelial’ regions included ductal epithelial cells and ducts, while ‘Stroma’ regions included collagen-rich stromal regions without any epithelial structures. ‘Outer’ regions were outside of the biopsied tissues. These were used for training and to test for the supervised machine-learning approach. We first trained the SegNet network using 50% of randomly selected images (38 images) for a total of 76 images. The remaining images were used for a validation test. Figure 3A and Appendix A show the segmentation results. It seemed that segmentation results showed good performance both for FA and PT images. Differences indicated by magenta or green in the images were not significantly different. For the test data, the differences became slightly larger compared to the training ones (Figure 3B). However, the absolute differences still remained small. In order to evaluate the segmentation performance quantitatively, we examined the total accuracy and the intersection of union (IoU) between the predicted and ground-truth images (Figure 3C). The total accuracy for the test image sets was 93.5%, and the IoU for those was 89.5%, indicating a high segmentation performance. To show the reproducibility of the machine learning performance, we again ran network training using another 50% of randomly selected images, keeping other parameters exactly the same as the previous ones. The results showed a high performance of segmentation, the total accuracy for test image sets was 93.8%, and the IoU for those was ~90.5% (Appendix A).

We further investigated performance results when the number of training images decreased. We examined the case in which 20% of randomly selected images (15 images) was used for training data. The predictive algorithm still showed good performance (the total accuracy ~91%, the IoU ~86%), indicating that a small training set was sufficient for accurate results (Appendix A).

### 2.3. Computer-Assisted Scoring Helps to Diagnose FA and PT Lesions

Based on the result of the image segmentation analysis, we performed a scoring of the epithelial to stromal area ratio. The ratio in PTs was higher than FAs, because the leaf-like architecture was reflected in the epithelial area, including the lumen. We evaluated this score for each image and calculated its statistics (average∓standard deviation) for both ground-truth and predicted image data (Figure 4A). Both data showed that the score for PT was higher than that for FA. Therefore, the score could be an index to classify fibroepithelial lesions. The scores calculated using the ground-truth and predicted data showed almost the same values, suggesting that AI-based segmentation using supervised image data sets could return accurate signatures for diagnosis.

We looked for another quantitative feature for fibroepithelial tumor differentiation. Carefully looking at the SHG images, stroma in FA samples emitted slightly stronger SHG signals than those in PT samples (Figure 1, Appendix A). To investigate these points, we quantified the SHG image intensities within the stromal regions (Figure 4B). The averaged signal intensities showed that the FA sections emitted stronger signals than the PT sections. The PSR-stained sections showed a similar extracellular collagen deposition; thus, this suggested different collagenous patterns between FA and PT lesions, demonstrating an advantage of using MPM for evaluating tissue samples. We combined the scores of the epithelial to stromal area ratio and the SHG signal intensity within the stromal area. Two-dimensional scatter plots showed a clear separation of FA and PT samples. Although these indicated that individual values of the epithelial to stromal area ratio showed some mixture of the two lesions (Figure 5A,B), the SHG signal intensity clearly differentiated the two lesions. To computationally confirm this point, we performed linear discriminant analyses (Appendix A). We performed the analyses for the original size (512 × 512 px) and the small size (128 × 128 px) image sets. The small size image sets were generated by dividing the original image into 4 × 4 blocks. From these images, 500 images were randomly selected (excluding the stromal area = 0 block) for both FAs and PTs. Then, these images were subjected to score calculations. The results showed a high accuracy of differentiation (Appendix A). The lines separating the two lesions were almost perpendicular to the axis of the SHG signal intensity, suggesting that the SHG intensity was highly accurate in diagnosing features for fibroepithelial lesions.

## 3. Discussion

FAs and PTs are fibroepithelial tumors, and consist of the proliferation of both epithelial and stromal elements. FAs are a concurrent proliferation of glandular and stromal elements. The stroma is usually hypocellular and may be fibrous, myxoid, or hyalinized. There is no stromal atypia and few mitotic activities. On the other hand, PTs are hypercellular fibroepithelial tumors characterized by an exaggerated stromal growth pattern with a leaf-like architecture. The leaf-like architecture is the elongated epithelial-lined clefts, resulting from stromal overgrowth just below the epithelium. The accurate evaluation of this morphological feature raises the possibility of the distinction of benign breast tumors. In this study, we highlighted this point with the help of MPM and a machine learning tool.

Label-free imaging using MPM enabled us to observe unstained samples using endogenous sources of non-linear signals and to diagnose several types of disorders. In breast tissues, the fibrosis assessment based on the SHG signal, which comes from collagen molecules upon two-photon excitation, has previously been investigated [30]. Changes in the collagen architecture in breast lesions could be observed. Therefore, SHG has been used to quantitatively characterize fibrillar collagen deposition. Furthermore, the strength of the SHG signal correlates with the molecular organization of living tissues. We actually obtained different signal levels of SHG intensity for FA and PT tissues, even though the histological staining results did not show clear differences between the two tissue images. This suggested that the two lesions show different ways of collagen deposition, and a different steric architecture of collagen fibrils. Further studies on these mechanical insights of how collagen accumulates in the fibroepithelial lesions are required. On the other hand, a strong native fluorescence was emitted in the breast tissues. This included NADH, flavins, and vitamins [19,20,21]. We used an excitation laser at a 950 nm wavelength for MPM image acquisition, in which the primary intracellular sources of fluorescence in liver tissues are NADH and flavins [15]. The NADH and flavins allowed us to visualize epithelial and stromal morphology, and to discuss histological characteristics. An advantage of using MPM is to estimate quantitative features, such as fluorescent and SHG signal intensities. Here, we evaluated SHG signal intensities which were not affected by photo-bleaching, such as fluorescence. In addition, these methods did not show inter-assay differences, such as staining variability, which led to descriptive and semi-quantitative evaluations arising from observer discrepancies.

We evaluated the SHG signal intensity and the epithelial to stromal area ratio. To proceed with the computer-assisted diagnosis method, we used SegNet, a deep convolutional neural-network-based image segmentation tool. SegNet returns a high accuracy of predicted image data. We first tried 50% of total images for the training data. This resulted in over 90% coincidence between the ground-truth and predicted data. Next, we reduced the number of images used for training to 20% of total images. This also showed a 90% accuracy. This means that larger numbers of images were not required, and suggests that the method can be applied easily to new images, once a reliable data set is constructed. The two types of scores were useful to differentiate FA and PT lesions. Scatter plots and a discrimination analysis revealed that the combination of the two scores is essential for individual classification. The SHG intensity reflected the molecular organization of collagens; thus, MPM has an essential role in diagnosing fibroepithelial lesions. Although deep learning costs computational demand, once a neural network capable of returning reliable results was constructed, the image segmentation applied to the test data was rapidly obtained. Thus, through fixing the acquisition conditions, we could realize a computer-assisted objective diagnostic method for detecting breast lesions.

## 4. Materials and Methods

### 4.1. Patient-Derived Samples

The specimens were obtained from 10 female patients who were diagnosed with breast fibroepithelial tumor by core needle biopsy (CNB) or vacuum-assisted biopsy (VAB). All patients underwent lumpectomy or mastectomy between January 2012 and July 2018, and were pathologically diagnosed with FA or PT.

### 4.2. Preparation of Tissue Sections

The biopsy samples were fixed with 10% neutral buffered formalin for 24 h at room temperature, and were subjected to embedding in paraffin. Sections 5 μm thick were cut and stained with hematoxylin–eosin (HE) and picro-sirius red (PSR). PSR staining was performed using a picro-sirius red stain kit (Polysciences, Inc., Warrington, PA, USA), which stains type I and type III collagens. Bright field images of the sections were acquired using a wide-field inverted microscope (All-in-one fluorescence microscope BZ-X700, Keyence, Inc., Osaka, Japan) with a 20× magnification objective lens (PlanFluor 20× NA:0.45, Nikon, Inc., Tokyo, Japan).

### 4.3. Image Acquisition by Multi-Photon Microscopy

We utilized upright MPM (A1R-MP, Nikon, Inc., Tokyo, Japan) equipped with a water immersion lens (CFI75 Apo 25 × W MP, NA:1.1, Nikon, Inc., Tokyo, Japan), and a Ti:sapphire laser oscillator system (MaiTai eHP, Spectra-Physics, Inc., Milpitas, CA, USA) for observing SHG and AF signals, as described previously [15,17]. For the detection of SHG and AF signals, we employed excitation wavelengths of 950 nm with emission filter sets, including (1) the dichroic mirror (DM) 495 nm and the short-pass filter 492 nm, (2) DM 560 nm and bandpass filter 525/50 nm (center wavelength/bandwidth), and (3) DM 662 nm and bandpass filter 617/73 nm. The field of view (FoV) of the single images was 0.5 mm × 0.5 mm, and the resolution was 512 × 512 pixel, (i.e., the pixel size was 1 μm). Larger FoV images (whole tissues and 1 mm × 1 mm FoV) were obtained by stitching the single images. The images were originally recorded as 12-bit gray level images, and were converted to 8-bit gray level images when analyzed computationally. For each patient sample, 6–12 regions were imaged and, in total, 33 and 43 images for FAs and PTs, respectively, were acquired.

### 4.4. Image Segmentation by SegNet

The automated image segmentation of MPM images was performed using SegNet, a deep convolutional neural network architecture for multi-class pixel-wise segmentation [31]. For this supervised learning-based image segmentation, we first prepared the training image data sets manually (ground truth). These data were a set of labeled (multi-level) images, which were composed of three kinds of regions, “Epithelial,” “Stroma,” and “Outer” regions. By comparing the HE- and PSR-stained section images, any pixels of MPM images were classified into the three categories using area selection tools in Fiji (Image J) software. “Epithelial” regions included ductal epithelial cells and ducts, while “Stroma” regions included collagen-rich stromal regions without any epithelial structures. “Outer” regions were outside of biopsied tissues. For training the SegNet network, 50% or 20% of randomly selected images of a total of 76 images were used, while the remaining images were used for the validation test. The learning parameters were as follows: momentum 0.9000, initial learn rate 0.0100, L2 regularization 0.0005, epoch number 5000, mini-batch size 4. The learning was performed by the stochastic gradient descent algorithm with momentum. For image augmentation of the network training, we used randomized pre-processing operations of image flip and translation (up to 20 pixels). In order to evaluate segmentation performance, we examined the total accuracy and the intersection of union (IoU) between the predicted and ground-truth images. The calculations for image segmentation were performed using the software MATLAB (MathWorks, Inc., Natick, MA, USA).

### 4.5. Statistical Analysis

The non-parametric statistical test was performed by the Kolmogorov–Smirnov test with a *p*-value < 0.05.

## 5. Conclusions

In order to establish a method for scoring fibroepithelial lesions in breast tissues, we used multi-photon excitation microscopy and computational image analyses. Deep-learning-based image segmentation was useful for differentiating epithelial and stromal regions in the lesions. We evaluated the potential utility of scoring methods for classifying fibroepithelial tumors in breast tissues. We showed that the combined features of the SHG intensity and epithelial to stromal area ratio accurately differentiated diseased tissue images. Therefore, the proposed method of computer-guided diagnosis would provide a promising approach for the morphological- and molecular-based diagnosis of breast tumors.

## Figures and Tables

**Figure 1 molecules-27-03340-f001:**
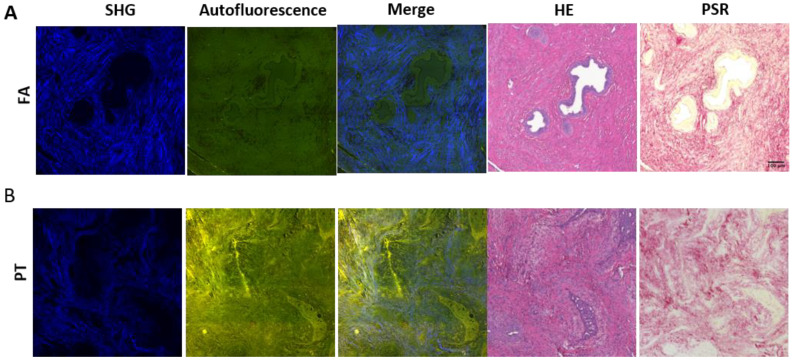
Image comparisons of the FA and PT lesions. Serial section images of multi-photon microscopy (MPM) and histological sections of HE and PSR staining for the FA (**A**) and PT (**B**) lesions. MPM images include the SHG signal, indicated in blue, and autofluorescence signal, indicated in green and red. PSR-stained collagen type I and III in red, and cell cytoplasm in light yellow. Scale bar, 100 μm.

**Figure 2 molecules-27-03340-f002:**
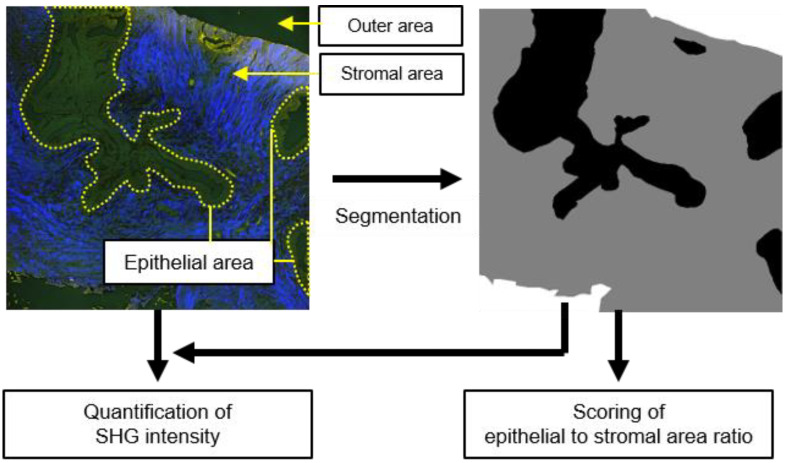
Schematic of quantification strategy for breast fibroepithelial lesions based on the multi-photon microscopy (MPM) image. Areas surrounded by yellow dotted lines denote lactiferous duct epithelia and lumens, while areas outside of those areas denote stroma. All images acquired by MPM were subjected to manual segmentation to construct ground-truth image sets for automated image analysis. Using ground-truth image sets as training image data, supervised machine learning of pixel-wise image segmentation was performed, which assigned all pixels to the epithelial, stromal, or outer areas. On the basis of the segmented image sets, measurement of SHG intensity within the stromal area and scoring lateral duct epithelial to stromal area were performed.

**Figure 3 molecules-27-03340-f003:**
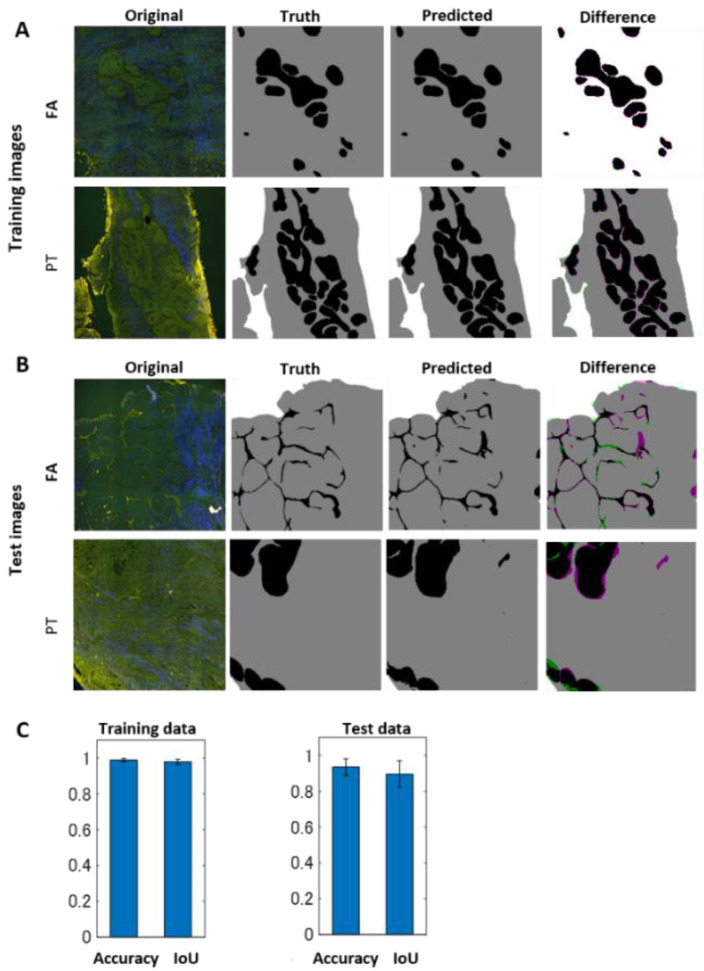
Results of image segmentation by a deep-learning-based framework, SegNet. (**A**) Results of training image sets. Original multi-photon microscopy images, ground-truth images, predicted images, and difference images are shown from left to right for both FA and PT images. Differences in images indicate FN areas as magenta and FP areas as green. (**B**) Results of test image sets. (**C**) Numerical evaluation of the segmentation results. The total accuracy between the ground-truth and predicted images and the weighted IoU, which indicates the area weighting sum of each IoU value, is shown for training and test data sets. These numerical values were evaluated for each image in training and test cases, and statistics such as mean and standard deviation were calculated. The bar denotes average; the error bar denotes standard deviation over the data calculated from image sets.

**Figure 4 molecules-27-03340-f004:**
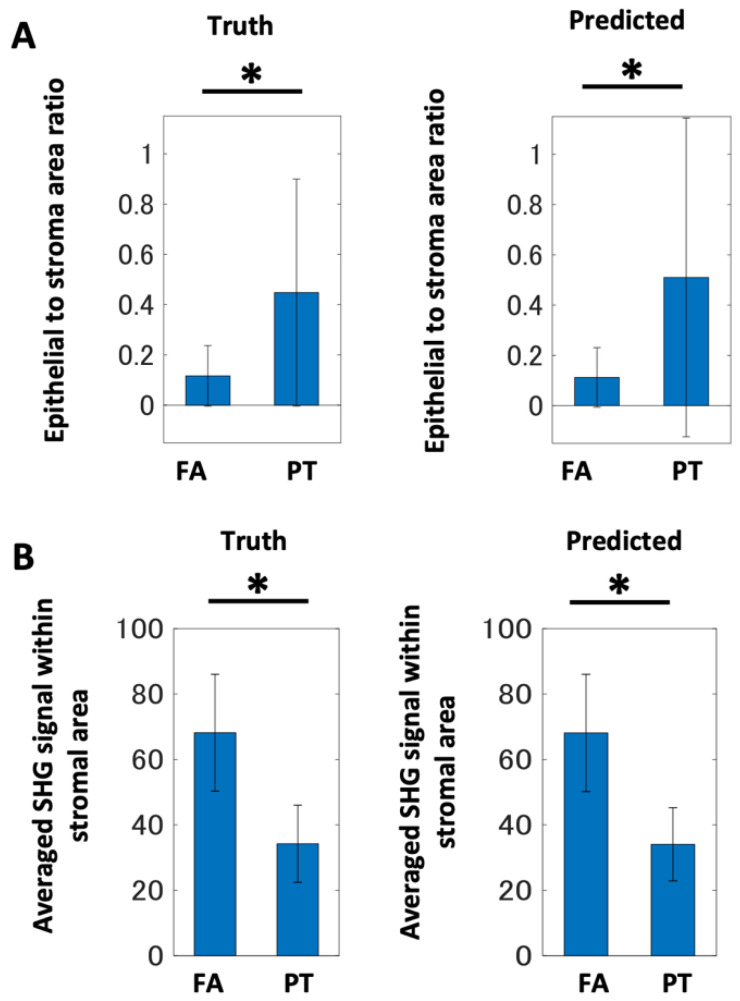
Quantification results of multi-photon microscopy images for breast fibroepithelial lesions. (**A**) Epithelial to stromal area ratio for FA and PT lesions. (**B**) Averaged SHG signal intensity within the stromal area for FA and PT lesions. These scores were evaluated for each image in training and test cases, and statistics such as mean and standard deviation were calculated. The bar denotes average; the error bar denotes standard deviation over the data calculated from image sets. Asterisks indicate statistical significance with the Kolmogorov–Smirnov test with a *p* < 0.05.

**Figure 5 molecules-27-03340-f005:**
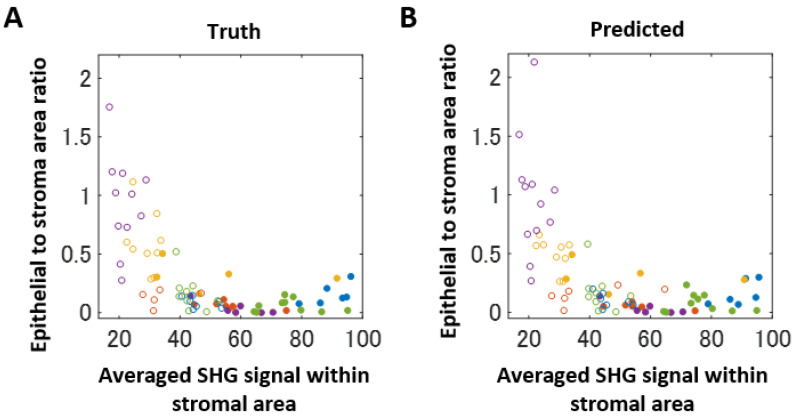
Scatter plots of the two quantification scores. (**A**) Scatter plot for the ground-truth data. (**B**) Scatter plot of the predicted data. The filled and open circles denote FA and PT data, respectively. The same color represents samples derived from the same patient.

**Table 1 molecules-27-03340-t001:** Summary of patient statistics.

Characteristic	Post-Operative Diagnosis	
	Fibroadenoma	Phyllodes
**No. of patients**	5	5
**Age** (median years)	38 (IQR; 27–41)	44 (IQR; 40–47)
**Median size on Imaging** (cm)	3.0 (IQR; 3.0–3.1)	2.9 (IQR; 1.4–3.5)
**Number of biopsy** (*n*) (min–max)	3 (2–4)	3 (3–6)
**Type of biopsy** (*n*)		
Core needle biopsy (14 gauge)	3	4
Vacuum-assisted breast biopsy (10 gauge)	2	1
**Pre-o****p****e****rative diagnosis** (*n*)		
Fibroadenoma	3	2
Phyllodes	0	3
Difficult to distinguish	2	0
**Histological type** (*n*)		
Benign		5
Borderline/malignant		0

## Data Availability

The data presented in this study are available in the Appendix A.

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
