# Peer review of "Computer-Aided Detection of Quantitative Signatures for Breast Fibroepithelial Tumors Using Label-Free Multi-Photon Imaging"

_molecules, 2022, doi:10.3390/molecules27103340_

Round 1

Reviewer 1 Report

A brief summary:

In this paper, the authors presented the classification of the Fibroadenomas (FAs) and phyllodes tumours (PTs) by using deep-learning methods on the tissue images obtained by multi-photon excited autofluorescence and second harmonic generation (SHG) method. 

Summary 

The study is oriented on a combined technique of in vitro imaging and computer-aided detection for differentiating fibroepithelial lesions, quantifying the collagen fibrils through both the second harmonic generation signals and multi-photon excited autofluorescence. This study provides a solution to the clinical management of PTs that differs from FAs due to the challenges of conventional techniques such as core needle biopsy diagnoses. Collectively, this study presents robust advantages to refining precise signatures for differentiating lesions through a machine learning-based approach, aiming to automate assessment. Finally, the proposed study will give us the opportunity to a high-throughput approach for evaluating precisely morphological- and molecular-based diagnoses of breast tumors.

General concept comments.

The manuscript is well-structured and clear.

Cited references are relevant and include appropriate self-citations.

Not only ref.24&25 I would like to see more self-citations which gave a snapshot of the evolution of research, especially technical parts (such as thesis related to presented SHG microscopy and statistical methods).

Based on “Therefore, although there are several predictive factors [3, 9-11], distinction between benign PTs and FAs still remains a difficult task, and hence stands as a diagnostic challenge to decide clinical management.” section: “still remains a difficult task” proposition can be supported by references.

In lines, 81-82 “visualization of molecules possessing non-centrosymmetric” might be “visualization of molecules possessing non-centrosymmetric nature”.

Figure 3 provides image and a deep learning-based framework comparisons of FA and PT lesions. However, it can be seen the orientation of the tissue segments are different. The same location can provide the clear conclusions since the clinical management of PTs differs from FAs is challenging.

In section 2.2 line 164 phrase “It has been partly reported that FA grows more in stromal regions than PT” needs a reference. 

The authors say in lines 165-166 “for establishing a quantitative criterion for differentiating FA and PT, we attempted to score the epithelial to stromal area ratio.” 

The manuscript is scientifically sound and the experimental design is appropriate to test the hypothesis. Besides, section 2.3, Figure 4 and Figure 5 must be revised. It is not clear to interpret the SD error bars in Fig.4A.  (for example see the Reference 24, Figure 4C of the same authors another work)

Ideally, the epithelial/stromal area ratios that will be selected from FA and PT might be equal and it’s not appropriate that the analysis of these data by using LDA. Nevertheless, it can be suited for generating the random images into the small area (100 x 100 px) that found at a single image (512 x 512 px)  and once obtained the ratios, LDA can be applied those small random images.

The authors expressed  in section 4.5 that “Non-parametric statistical test was performed by the Kolmogorov-Smirnov test with 427 a p-value < 0.05” however did not highlighted on the text, used where and which context (for example see the Reference 24, Figure 4C of the same authors another work)

Summary, except for trivial items this manuscript is generally clear and closely related to the relevant field and has presented in a well-structured manner. 1/3 of cited references are recent publications and no observed an excessive number of self-citations. Just as the manuscript is scientifically sound and the design of the experiment is testable for the hypothesis, the results of the manuscript have been reproducible based on the details provided in the method section. Even though most figures and provided data have been presented properly, the data/figure section can be improved to clarify the hypothesis. Holistically, the whole manuscript will be easy to interpret and understand once the data have improved. Additionally, provided data is interpretable properly and consistently throughout the manuscript mostly. The conclusion is consistent with the provided evidence and presented arguments.

Author Response

We would like to express our sincerest gratitude for your valuable comments. Based on your comments, we conducted additional analysis, and accordingly revised the text and figures. We believe your comments have enhanced the importance of this work. The modified sentences are highlighted as red-colored text in the manuscript.

The manuscript is well-structured and clear.

Cited references are relevant and include appropriate self-citations.

Not only ref.24&25 I would like to see more self-citations which gave a snapshot of the evolution of research, especially technical parts (such as thesis related to presented SHG microscopy and statistical methods).

Response: Thank you for the comments. We added references regarding application of multi-photon techniques to disease assessment, please see line74. According to this change, the reference number was also changed.

Based on “Therefore, although there are several predictive factors [3, 9-11], distinction between benign PTs and FAs still remains a difficult task, and hence stands as a diagnostic challenge to decide clinical management.” section: “still remains a difficult task” proposition can be supported by references.

Response: We agree with your recommendation and have revised “Therefore, although there are several predictive factors [3, 9-11], these are still insufficient to perform distinction between benign PTs and FAs, and hence standss as a diagnostic challenge to decide clinical management [4-6].”n line 69 in the revised manuscript. As suggested, references were added to support the proposition.

In lines, 81-82 “visualization of molecules possessing non-centrosymmetric” might be “visualization of molecules possessing non-centrosymmetric nature”.

Response: The word was corrected as suggested in line 83 in the revised manuscript.

Figure 3 provides image and a deep learning-based framework comparisons of FA and PT lesions. However, it can be seen the orientation of the tissue segments are different. The same location can provide the clear conclusions since the clinical management of PTs differs from FAs is challenging.

Response: The images we have shown in Fig3 are typical tissue regions for FA and PT. The left most one is original AF/SHG image, and other images are ground truth, the Segnet-predicted, and difference between these images, in which the regions are exactly same as the original AF/SHG image. However, the number of samples shown is small. Thus, for further help of clear understanding, we added other segmentation results of FA and PT as a new Supplementary Fig4.

In section 2.2 ,line 164 phrase “It has been partly reported that FA grows more in stromal regions than PT” needs a reference.

Response: This statement “It has been partly reported that FA grows more in stromal regions than PT” is incorrect, thus this is corrected to “It has been partly reported that PT grows more in stromal regions than FA”. Besides, the reference which support this statement [2] is added. Please see line 155 in the revised manuscript.

The authors say in lines 165-166 “for establishing a quantitative criterion for differentiating FA and PT, we attempted to score the epithelial to stromal area ratio.” The manuscript is scientifically sound and the experimental design is appropriate to test the hypothesis. Besides, section 2.3, Figure 4 and Figure 5 must be revised. It is not clear to interpret the SD error bars in Fig.4A.  (for example see the Reference 24, Figure 4C of the same authors another work)

Response: For avoidance of confusion, we added an explanation of this calculation in Figure legends. Please see legends of Fig3, Fig4, and Supplementary Fig5 and Fig6.

Ideally, the epithelial/stromal area ratios that will be selected from FA and PT might be equal and it’s not appropriate that the analysis of these data by using LDA. Nevertheless, it can be suited for generating the random images into the small area (100 x 100 px) that found at a single image (512 x 512 px) and once obtained the ratios, LDA can be applied those small random images.

Response: We consider this comment suggest that the number of images subjected to the LDA analysis should be equal between FAs and PTs and also the small size random images set should be used. Thus, we performed additional analysis. We first generated smaller size image set (128 x 128 px) by simply dividing the original (512 x 512 px) image into 4x4 blocks. Then, from these images, 500 random image sets were selected ((excluding the stromal area = 0 block) for both FA and PT and these were subjected to calculation of the epithelial to stromal area ratio and SHG signal within stromal area. With these numerical values, LDA analysis was performed. We obtained the result which is similar to the one with the original image sets. To reflect these to the manuscript, we added data to Supplementary Fig7 and modified the text section 2.3 line 240.

The authors expressed in section 4.5 that “Non-parametric statistical test was performed by the Kolmogorov-Smirnov test with 427 a p-value < 0.05” however did not highlighted on the text, used where and which context (for example see the Reference 24, Figure 4C of the same authors another work)

Response: We added the sentence which explains the statistical test in Fig 4 legends.

Summary, except for trivial items this manuscript is generally clear and closely related to the relevant field and has presented in a well-structured manner. 1/3 of cited references are recent publications and no observed an excessive number of self-citations. Just as the manuscript is scientifically sound and the design of the experiment is testable for the hypothesis, the results of the manuscript have been reproducible based on the details provided in the method section. Even though most figures and provided data have been presented properly, the data/figure section can be improved to clarify the hypothesis. Holistically, the whole manuscript will be easy to interpret and understand once the data have improved. Additionally, provided data is interpretable properly and consistently throughout the manuscript mostly. The conclusion is consistent with the provided evidence and presented arguments.

Response: Thank you so much for encouraging comments. 

Reviewer 2 Report

The manuscript shows the result from a convolutional neural network trained by multi-photon microscopy images to classify breast fibroepithelial tumors (fibroadenomas (FA) and phyllodes tumors (PT)) by differentiating epithelial and stromal regions. The average collagen from second harmonic generation signal within the stromal area show clear difference between FA and PT images. However, the manuscript is not organized well and below should be revised.

  1. Data from 5 patients were analyzed. The number of patients is too small. Also, number of images for training is very small. There is a possibility that the training and test data are similar so the test result is supposed to be good. However, it may not work for other patient data. To confirm the performance of the network, test the network using data from other patients also.
  2. As low number of data was used for training, consider augmentation to improve the network.
  3. Image resolution for all figures is too low. Please insert images in higher resolution. Alignment (e.g., Fig 1) and figure cations should be checked (e.g. Fig 4).
  4. Both HE and PSR stained images are used as a reference to generate ground-truth images? Authors explained why PSR staining is necessary but not for HE. Please describe why both staining is required.
  5. Please add more detail about the experiment (e.g., image sizes, how to cut the image, etc).
  6. Please add more detail of the network training (e.g., hyperparameters).
  7. Section 4.5 statistical analysis, where is the result from this analysis? Why this method was chosen?
  8. The variation of epithelial to stroma area ratio is too high to ensure the difference between the FA and PT images (Figure 4A).
  9. Although the scatter plots (Figure 5) show the separation of FA and PT samples, separation was clearer from the SHG signal within stromal area than from the epithelial to stroma area ratio. Can this be improved?

Author Response

We would like to express our sincere appreciation for your thoughtful comments. As per your comments, we conducted additional analysis and accordingly revised the manuscript. We believe that these changes enhanced the importance of our work. Red text in the revised manuscript denotes modification.

  1. Data from 5 patients were analyzed. The number of patients is too small. Also, number of images for training is very small. There is a possibility that the training and test data are similar so the test result is supposed to be good. However, it may not work for other patient data. To confirm the performance of the network, test the network using data from other patients also.

Response: Thank you for your insightful comments. PT is very rare disease:0.3-1% of all primary tumor of breast. PT sometime exceed 4-5cm in size, in such case, clinical diagnosis is relatively easy. However, when tumor is less than 3-4cm in size, it may be difficult to distinguish between PT and FA. Therefore, we analyzed tumor of similar size and the number of samples was small. Also, as commented by the reviewer, to confirm the reproducibility/robustness of the training, it is important to test using other image data sets. Thus, we additionally performed a network training using another 50% of randomly selected images, keeping other parameters exactly same as the previous one. The results are shown in Supplementary Fig5 and the section 2.2, line 177 in the revised manuscript. The result also showed high performance of segmentation, the total accuracy for test image sets is 93.8%, and the IoU for those is ~90.5%. Besides, we have already performed training using 20% of randomly selected images, which still showed good segmentation performance (the total accuracy ~91%, the IoU ~86%). Together with these data, although the number of training data set is small, the segmentation framework may return accurate and robust performance.

  1. As low number of data was used for training, consider augmentation to improve the network.

Response: As reviewer commented, augmentation of the training data is important to ensure the network versatility such as preventing the network from overfitting. For this, we used randomized preprocessing operations of image reflection and translation (20 px) of the training images. These information and detailed parameters for network training were added in the section 4.4 line 359 in Materials and Methods.

  1. Image resolution for all figures is too low. Please insert images in higher resolution. Alignment (e.g., Fig 1) and figure cations should be checked (e.g. Fig 4).

Response: We exchanged figures with higher resolution ones.

  1. Both HE and PSR stained images are used as a reference to generate ground-truth images? Authors explained why PSR staining is necessary but not for HE. Please describe why both staining is required.

Response: HE staining is necessary to find the regions of ductal epithelium and stromal area. Especially, cell nuclei stained in the HE slices are useful for recognizing ductal epithelia and lumens. Thus, to reflect this point, we slightly modified the text page line 125-136 as “On the places corresponding to the epithelia, cell nuclei were stained in the HE slices, indicating that these included mammary duct epithelia and lumens.”

  1. Please add more detail about the experiment (e.g., image sizes, how to cut the image, etc).

Response: Microscope and its conditions for image acquisition were described in the section 4.3 in Materials and Methods. Briefly, the originally acquired image size is 512x512 with the resolution 1um/pixel. The field of view (FoV) of the single images was 0.5 mm × 0.5 mm. 6-12 regions for each patient were selected.

  1. Please add more detail of the network training (e.g., hyperparameters).

Response: Detailed parameter values for network training were added in the section 4.4 line 359 in Materials and Methods.

  1. Section 4.5 statistical analysis, where is the result from this analysis? Why this method was chosen?

Response: We added the sentence which explains the statistical test in Fig4 legends. We chose this nonparametric statistical test because this can be used without assumption that data obeys parametric statistics.

  1. The variation of epithelial to stroma area ratio is too high to ensure the difference between the FA and PT images (Figure 4A).

Response: We appreciate this comment. This is an important point for our research. As pointed out, standard deviation was quite high in the epithelial to stromal ratio (Fig 4A), although this showed statistical significance. The reason for this can be understood by seeing scatter plots of individual data for two scores, the epithelial to stromal area ratio and SHG signal within stromal area (Fig5). Along with the ratio axis (y), data are widely spread and some mixtures of open (FA) and filled circles (PT) are recognized. While the SHG signal clearly separate the data. With these data, the epithelial to stromal ratio can be a rough differentiating factor of the breast lesions, and the SHG signal intensity within the stromal area provides higher accurate FA and PT classifications. Thus, as described in the section 5, our conclusion is that the combined features of SHG intensity and epithelial and stromal area ratio accurately differentiated diseased tissue images. Additionally, to improve the data visibility, we modified the axis range of Figure 4A.

  1. Although the scatter plots (Figure 5) show the separation of FA and PT samples, separation was clearer from the SHG signal within stromal area than from the epithelial to stroma area ratio. Can this be improved?

Response: The stroma of the benign PT is usually more cellular than FA, but, due to stromal heterogeneity, it is difficult to distinguish both PA and FA in core needle biopsy. We thought the results of the epithelial to stromal area ratio reflected the stromal heterogeneity. Thus, as described in the response for 8 above, the epithelial to stromal ratio can be a rough differentiating factor of the breast lesions, and the SHG signal intensity within the stromal area provides higher accurate FA and PT classifications. We believe that the combined features of SHG intensity and epithelial and stromal area ratio accurately differentiated diseased tissue images.

Round 2

Reviewer 2 Report

Comment #2, please use more common term (reflection -> flip).  translation (px): is this in pixel? is it simply 20 pixels or up to 20 pixels?

Comment #3, Fig 4 caption is not correct. Please match with the figures.

Author Response

Reviewer#2

We thank you for your thoughtful suggestions and insights.

Yellow highlight in the revised manuscript denotes modification.

***************************************************************************************************

Comment #2, please use more common term (reflection -> flip).  translation (px): is this in pixel? is it simply 20 pixels or up to 20 pixels?

Response: We agree with your recommendation.

We have revised “reflection” to “flip” and “px” to “pixels” in the section4.4 revised manuscript. The translation is up to 20 px, so we revised the text in that way.

Comment #3, Fig 4 caption is not correct. Please match with the figures.

Response: Thank you for pointing this out. We modified the caption so as to match the Figure as (A) Epithelial to stromal area ratio for FA and PT lesions. (B) Averaged SHG signal intensity within the stromal area for FA and PT lesions.

***************************************************************************************************